# Autophagy across tissues of aging mice

Julian M. Carosi[1,2], Alexis Martin[1], Leanne K. Hein[1], Sofia Hassiotis[1,3], Kathryn J. Hattersley[1], Bradley J. Turner[4], Célia Fourrier[1,5], Julien Bensalem[1,5], Timothy J. Sargeant[1,6]*

1 Lysosomal Health in Ageing, Lifelong Health Theme, South Australian Health and Medical Research Institute (SAHMRI), Adelaide, South Australia, Australia, 2 Faculty of Sciences, School of Biological Sciences, Engineering and Technology, The University of Adelaide, Adelaide, South Australia, Australia, 3 Future Industries Institute, University of South Australia, Adelaide, South Australia, Australia, 4 The Florey Institute of Neuroscience and Mental Health, University of Melbourne, Parkville, Victoria, Australia, 5 Adelaide Medical School, The University of Adelaide, Adelaide, South Australia, Australia, 6 Faculty of Health and Medical Sciences, The University of Adelaide, Adelaide, South Australia, Australia

* tim.sargeant@sahmri.com

## Abstract

Autophagy is a 'waste-disposal' pathway that protects against age-related pathology. It is widely accepted that autophagy declines with age, yet the role that sex and diet-related obesity play during aging remain unknown. Here, we present the most comprehensive *in vivo* study of autophagic flux to date. We employed transgenic mice overexpressing tandem-florescent LC3B (RFP-GFP-LC3B) to measure autophagic flux in the blood (PBMCs), heart, and motor cortex neurons of aging mice that were fed regular chow or a high-fat diet for 6-, 12- or 18-months. In male mice, aging decreased autophagic flux in the heart, increased it in the blood, and had no effect in motor cortex neurons. Age-dependent changes autophagic flux were less pronounced in female mice. High-fat diet influenced autophagic flux in the blood and heart of male but not female mice. Overall, we uncovered sexual dimorphisms that underpin how autophagy changes with age across different tissues and in response to a high-fat diet.

## Introduction

Autophagy captures cytoplasmic components (e.g., damaged organelles, protein aggregates, invading pathogens) within autophagosomes and delivers them to lysosomes for destruction and recycling of their building blocks [1]. This process homeostatically remodels the intracellular environment in response to diverse stimuli [1], and at an organismal level is required for proper fetal development [2,3], adaptation to starvation [4,5], innate immune function [6,7], and prevention of age-related pathologies [8–11].

During autophagy, ATG8 homolog proteins (LC3A, -B, -C and GABARAP, -L1 and -L2) are cleaved immediately after a conserved glycine residue in their C-terminus

**Data availability statement:** All relevant data are within the manuscript and its Supporting Information files.

**Funding:** This investigation was supported by Lysosomal Health in Ageing at SAHMRI, and an Ideas Grant from the National Health and Medical Research Council (GNT2002608) awarded to JB and TJS. JMC is supported by an EMCR Fellowship from The Hospital Research Foundation Group (2022-CF-EMCR-007). The authors acknowledge Microscopy Australia (ROR: 042mm0k03) resources at the Future Industries Institute, University of South Australia, enabled by NCRIS. The funders had no role in study design, data collection and analysis, decision to publish, or preparation of the manuscript.

**Competing interests:** We declare no conflicts of interest.

**Abbreviations:** CASM, conjugation of ATG8s to single membranes; PBMCs, peripheral blood mononuclear cells; PBS, phosphate-buffered saline; tf-LC3B, tandem fluorescent LC3B/RFP-GFP-LC3B.

[12], which enables their direct conjugation to lipids on the growing phagophore (LC3B-II refers to its lipidated form) [13,14]. Here ATG8s serve important roles in phagophore maturation [15], the selective capture of substrates (via binding to autophagy cargo receptors) [16] and autophagosome-lysosome fusion [15]. As such, ATG8s are themselves destroyed by the lysosome in an autophagy-dependent manner [17]. LC3B-II levels are a reliable indicator of autophagosome/autolysosome abundance [17,18]. However, 'snapshot' measurements of LC3B-II alone cannot provide information about the rate at which autophagy is occurring (i.e., autophagic flux) [17,18]. Autophagic flux is simple to analyze in cultured cells via measurement of LC3B-II turnover by lysosomes in the presence of lysosome inhibitors (such as bafilomycin, chloroquine, leupeptin) [18], or using LC3B-based probes that report delivery of autophagosomes to lysosomes [19,20], or their lysosome-dependent turnover in the absence of lysosome inhibitors [21].

Critically, measurement of autophagic flux *in vivo* remains challenging. Although it is possible to measure LC3B-II turnover in mice exposed to lysosome inhibitors (such as chloroquine, colchicine, and leupeptin), their tissue distribution is variable and they do not always penetrate the blood-brain barrier [22–27]. Further, because different mice are treated with or without lysosome inhibitors this approach remains a crude estimate of autophagic flux. Moreover, transgenic mice that overexpress GFP-LC3B do not report autophagic flux because GFP fluorescence is lost within acidic autolysosomes, which makes it challenging to uncouple degradation and synthesis of the reporter [28]. pH-sensitive fluorescence allows LC3B-based reporters to track autophagosomes (neutral pH) and autolysosomes (acidic pH), providing a measure of autophagic flux *in vivo* by comparing their abundances (e.g., tandem fluorescent LC3B [tf-LC3B; RFP-GFP-LC3B] [29,30] and mKeima-LC3B [31]), or autophagosome degradation relative to synthesis (e.g., GFP-LC3B-RFP-LC3BΔG) [20].

Autophagy limits biological aging and extends lifespan [8–10,32], and it is widely accepted that autophagy declines with age [33]. Indeed, while the degradative capacity of lysosomes is reduced with age in *C. elegans* [34], it remains unclear whether autophagy is affected [35,36]. In rodents, basal autophagy remains stable in the liver except in extreme old age, while stress-induced proteolysis declines significantly from midlife [37–39]. Autophagic flux was shown to decrease with age in various tissues like liver, heart, kidney, and skeletal muscle, but not white adipose tissue where it increased with age [9,40–42]. Similarly, humans at risk for developing type-2 diabetes show a mild age-dependent increase in autophagic flux within their peripheral blood mononuclear cells (PBMCs) [43].

Aging, diet-related obesity, and sex increase the risk of various age-related diseases affecting the brain and heart. However, it remains unclear if and how these factors interact to influence autophagic flux *in vivo*. Additionally, since PBMCs are useful for measuring autophagic flux in humans [43,44], it is crucial to correlate their flux with that in disease-relevant tissues (e.g., brain and heart). Here, we present the most systematic *in vivo* exploration of autophagic flux to date, revealing how autophagic flux varies across tissues with age and the influence of a high-fat diet, highlighting the differences between sexes.

## Results

### Measurement of autophagic flux *in vivo* using tf-LC3B

To monitor autophagic flux in vivo we employed transgenic mice with tissue-wide overexpression of tf-LC3B [29]. After its cleavage and lipidation, tf-LC3B is incorporated into the phagophore and eventually autophagosome membranes [12–14,19,29,30]. Following autophagosome-lysosome fusion, GFP but not RFP signal from tf-LC3B is quenched by the acidic pH within the autolysosome [19,29,30]. tf-LC3 can therefore distinguish autophagosomes (GFP+ and RFP+; i.e., GFP+RFP puncta) from autolysosomes (GFP− and RFP+; i.e., RFP-only puncta), which can be used to estimate auto-phagic flux (**Fig 1A**) [19,29,30]: autophagic flux is high when the number of autolysosomes exceeds that of autophago-somes, and is low when the opposite occurs (Fig 1B) [19,29,30]. A high GFP:RFP ratio indicates that autophagic flux is low, whereas a low GFP:RFP ratio indicates that autophagic flux is high (**Fig 1B**). Using this approach, we investigated how autophagic flux changed with aging and high-fat diet at 6-, 12-, and 18-months (diet from 2-months of age) for each sex across several tissues including blood (PBMCs) via flow cytometry, or the heart and brain (motor cortex neurons) via confocal imaging (**Fig 1C**).

We first measured several physiological and metabolic factors to assess how they change with aging and diet for each sex. Mice fed a high-fat diet gained weight quicker than those fed regular chow (S1A Fig). Prior to humane killing, mice were fasted for 6 h prior to sacrifice to reveal blood glucose-related phenotypes on high-fat diet [45]. After 6- and 12-months of age, males but not females were sensitive to high-fat diet-induced changes in their fasting blood glucose levels (S1B and S1C Fig). Moreover, mice fed a high-fat diet displayed a higher incidence of developing liver tumors with age (S1D and S1E Fig).

### Autophagic flux increased in an age-dependent manner in the blood

Following extraction from blood, PBMCs from tf-LC3B-expressing mice were analyzed by flow cytometry. We confirmed the presence of monocytes (F4/80+), as well as T- (CD3+), and B lymphocytes (CD19+) in the population gated as PBMCs based on size (FSC-A) and granularity (SSC-A) (S2A Fig). In female mice, autophagic flux in PBMCs was not impacted by a high-fat diet (Fig 2A). Analysis of main-effects indicated that autophagic flux changed with age, but this was difficult to interpret (Fig 2A). Autophagic flux in the PBMCs of male mice strongly increased in an age-dependent manner (Fig 2B). Male mice fed a high-fat diet displayed slightly higher autophagic flux than those fed regular chow at 6-months of age, but this diet effect was not apparent at later time points (Fig 2B). Therefore, autophagic flux in PBMCs increased with age, but only young males were sensitive to high-fat diet-induced changes.

### Autophagic flux decreased in an age-dependent manner in the heart

Next, using confocal microscopy we analyzed how autophagy varied in the heart. In female mice, analysis of main-effects indicated that autophagic flux declined with age, but was not influenced by high-fat diet (Figs 3A,3B,S3A and S3B). In male mice on a regular chow diet, autophagic flux dropped between 6- and 12-months and continued to decrease by 18-months (Figs 3C,3D,S3C and S3D). However, in male mice on a high-fat diet, autophagic flux dropped between 6- and 12-months but remained unchanged between 12- and 18-months of age (Figs 3C,3D,S3C and S3D). By 18 months, male mice fed a high-fat diet had higher autophagic flux than those fed regular chow (Figs 3C,3D,S3C and S3D). Therefore, an age-dependent decrease in autophagic flux in the heart is modified by high-fat diet in males but not females (Fig 3A–D). These changes do not appear to correlate with expression levels of key autophagy machinery (S4A–S4D Fig).

### High-fat diet does not alter autophagic flux in motor cortex neurons

The brain is comprised of heterogeneous cell types including neurons, microglia, astroglia, and oligodendrocytes. Because paraffin embedded brain tissue for immunostaining dulled GFP and RFP signals (S5A Fig), cell-specific markers could

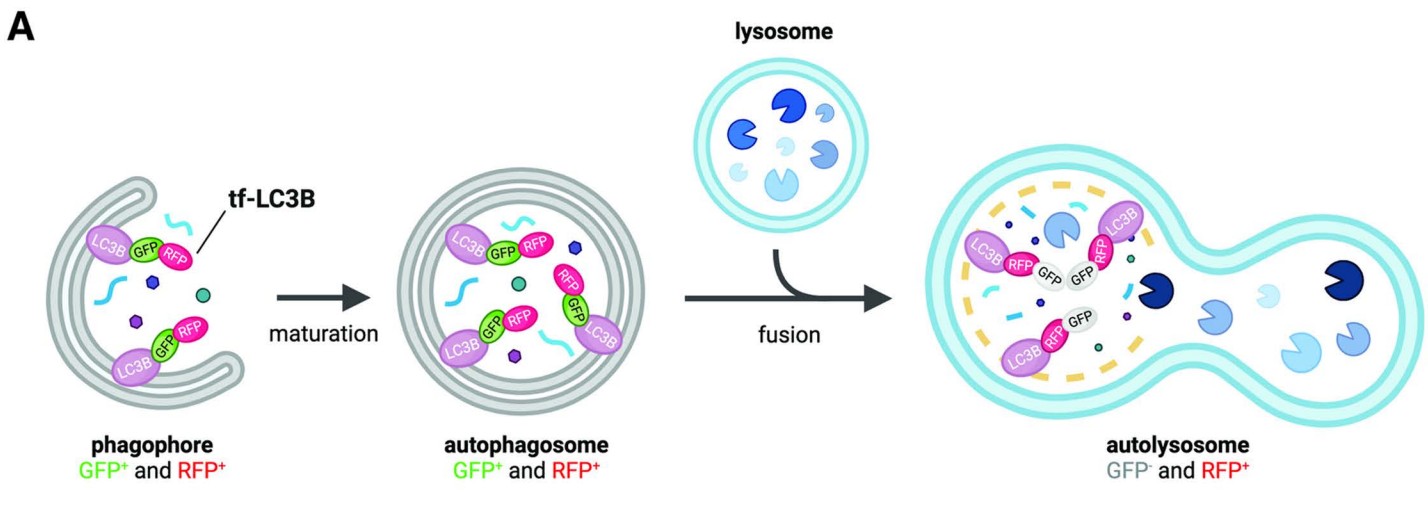

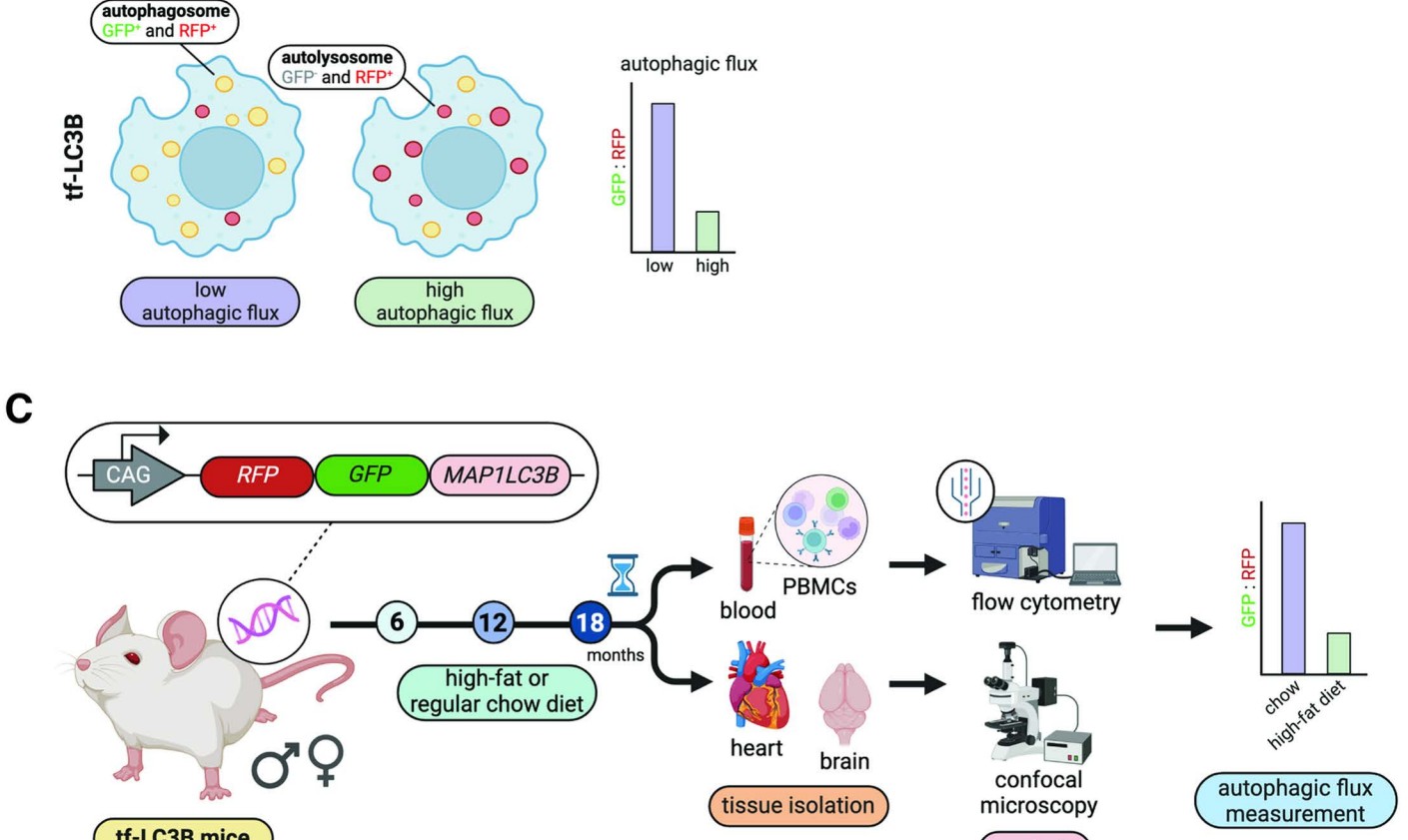

**Fig 1. Measurement of autophagic flux *in vivo* using tf-LC3B.** A. Diagram illustrating how tf-LC3B is used to measure autophagic flux in mouse tissues. B. Diagram illustrating how to interpret autophagic flux as indicated by the tf-LC3B probe. C. Summary of the experimental workflow for this study.

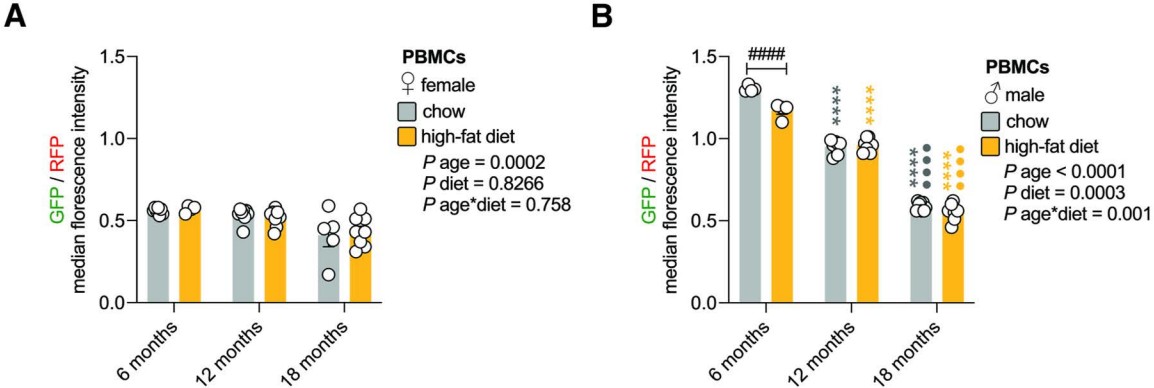

**Fig 2. Autophagic flux increased in an age-dependent manner in the blood.** A. Autophagic flux in female PBMCs. PBMCs extracted from female tf-LC3B mice that were fed regular chow or a high-fat diet for 6-, 12-, and 18-months were analyzed via flow cytometry. Values are GFP/RFP median fluorescence intensity ± sem for n = 4-8 female mice/timepoint (2-way ANOVA). B. Autophagic flux in male PBMCs. PBMCs extracted from male tf-LC3B mice that were fed regular chow or a high-fat diet for 6-, 12-, and 18-months were analyzed via flow cytometry. Values are GFP/RFP median fluorescence intensity ± sem for n = 4-8 male mice/timepoint (2-way ANOVA with Tukey's multiple comparisons test). Statistically significant p values comparing diet effects (#); age effects within each diet (coloured) compared to 6-month time point (*) or comparing 12- and 18-month time points (•).

not be used to determine how autophagy behaved in each cell type. Instead, we exploited unique morphological features of DAPI-stained nuclei from NeuN+ neurons (i.e., their large size, circular shape, unfilled/ "open" lumen and nucleoli-like inclusions) as a proxy to distinguish neurons in the motor cortex from other cell types such as microglia (IBA1+) or astroglia (GFAP+) (S5B and S5C Fig). Indeed, these characteristics could be reliably employed by a blinded experimenter to distinguish NeuN+ neurons exclusively from their DAPI-stained nuclei morphology, as indicated by receiver operating characteristic curve analysis (AUC = 1) (S5B and S5C Fig). Moreover, because lipofuscin throughout the brain exhibits broad autofluorescence, we excluded GFP+ or RFP+ puncta that appeared in the DAPI channel from our analysis (S5D and S5E Fig). In female mice autophagic flux and SQSTM1/p62-puncta in motor cortex neurons did not change in response to a high-fat diet (Figs 4A,4B,S6A,S6B,S7A and S7B). Analysis of main-effects indicated that autophagic flux changed with age, but this was hard to interpret (Figs 4A,4B,S6A and S6B). In male mice autophagic flux and SQSTM1/p62-puncta in motor cortex neurons did not change in response to diet or age (Figs 4C,4D,S6C,S6D,S7C and S7D). Therefore, autophagic flux in motor cortex neurons remains relatively stable with aging and in response to a high-fat diet.

## Discussion

Our systematic exploration of autophagic flux *in vivo* demonstrates how it changed with age, high-fat diet, and tissue for each sex. In male mice, aging and high-fat diet both modified autophagic flux in the blood (i.e., PBMCs) and heart, but not brain (motor cortex neurons). In female mice, age-dependent changes autophagic flux in motor cortex neurons were complex but statistically significant. In contrast to males, aging exerted milder effects on autophagic flux in the blood and heart of female mice, and was insensitive to dietary modification. This is consistent with the notion that females are protected against several metabolic and physiological consequences of a high-fat diet [46,47]. Although autophagic flux – as an absolute measure – was largely comparable between females and males in the brain and heart, females appeared to have higher autophagic flux in the blood. Autophagy has been reported to be sexually dimorphic in some contexts [48–50], and one plausible explanation for this is that the expression of lysosomal genes and their degradative functions are coordinated by sex hormones and their receptors [51]. Tissue-specific differences in autophagic flux likely reflect differences in their *in vivo* environment [28], sensitivity to stimuli [28], or expression and stoichiometry of components within [52] or upstream of the autophagy pathway [53].

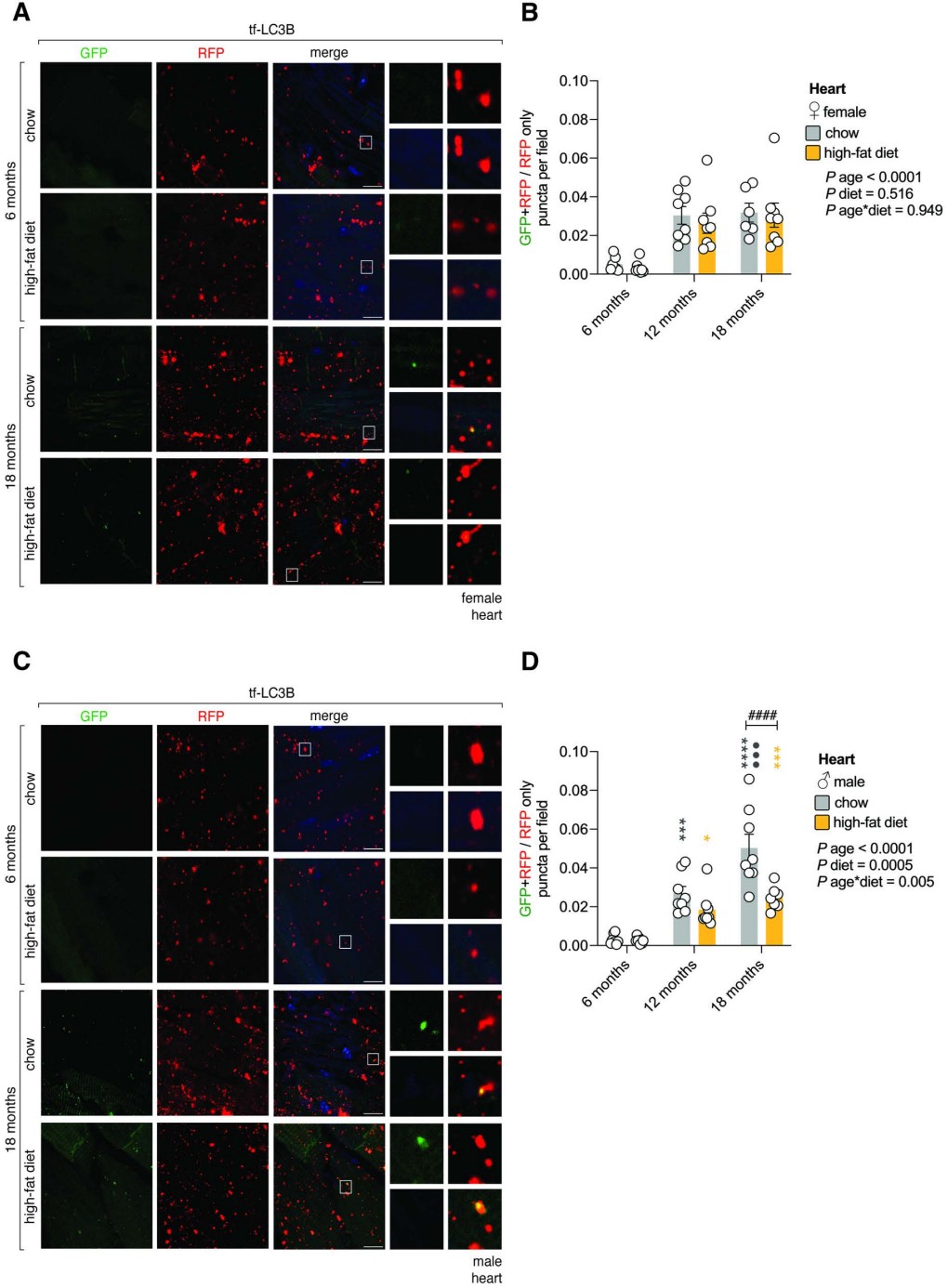

**Fig 3. Autophagic flux decreased in an age-dependent manner in the heart.** A. Autophagic flux in the female heart. Confocal images of the heart of female tf-LC3B mice that were fed regular chow or a high-fat diet for 6- and 18-months (12-month timepoint not shown). Scale bar: 10 μm. Inset: 10× magnification. Blue on merge: DAPI. B. Quantification of autophagic flux in the female heart. Values are GFP+RFP/RFP only puncta per field±sem for n=6-8 female mice/timepoint (2-way ANOVA). C. Autophagic flux in the male heart. Confocal images of the heart of male tf-LC3B mice that were fed regular chow or a high-fat diet for 6- and 18-months (12-month timepoint not shown). Scale bar: 10 μm. Inset: 10× magnification. Blue on merge: DAPI. D. Quantification of autophagic flux in the male heart. Values are GFP+RFP/RFP only puncta per field±sem for n=7-8 male mice/timepoint (2-way ANOVA with Tukey's multiple comparisons test). Statistically significant p values comparing diet effects (#); age effects within each diet (coloured) compared to 6-month time point (*) or comparing 12- and 18-month time points (•).

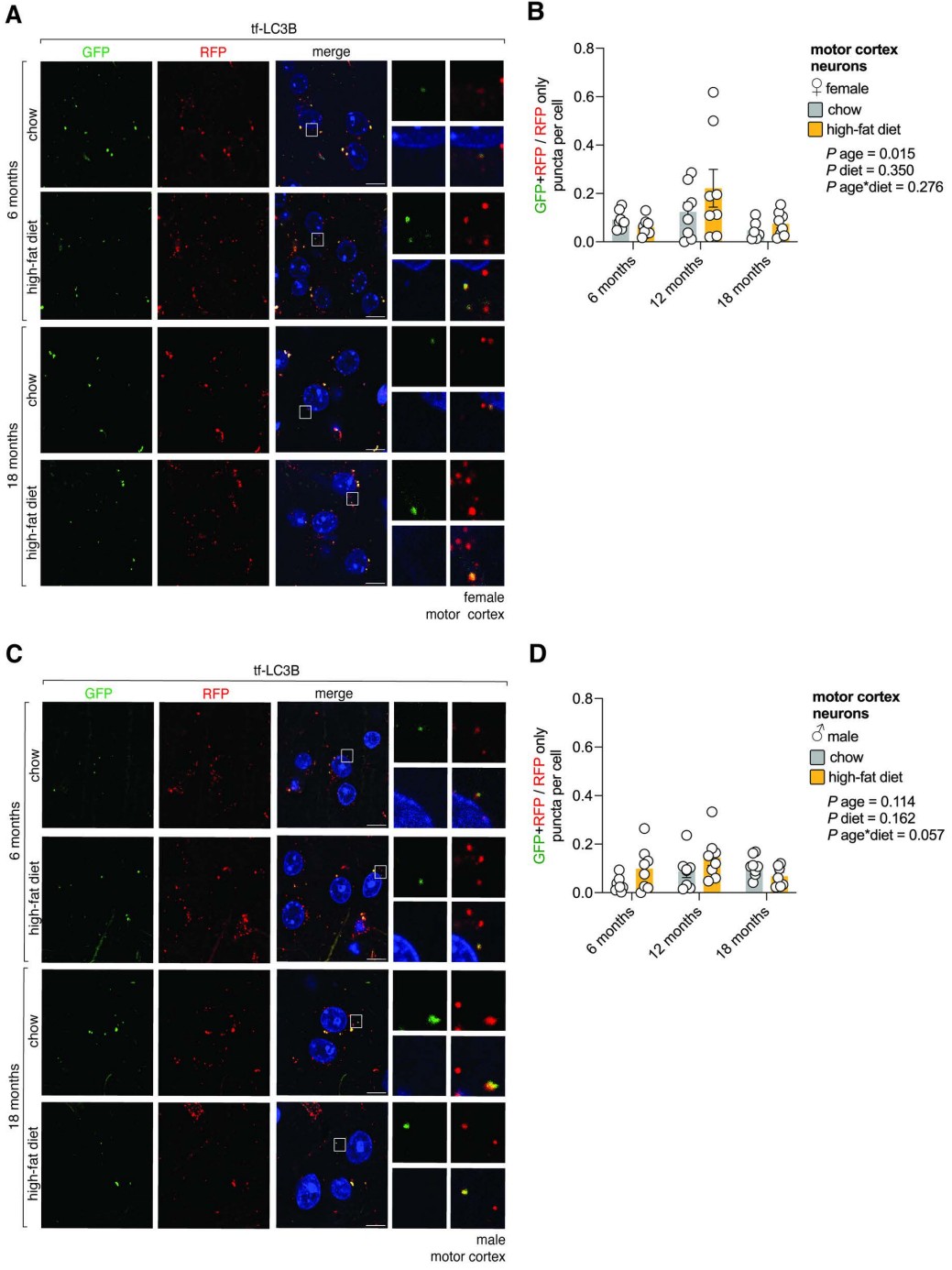

**Fig 4. High-fat diet does not alter autophagic flux in motor cortex neurons.** A. Autophagic flux in female motor cortex neurons. Confocal images of the motor cortex of female tf-LC3B mice that were fed regular chow or a high-fat diet for 6- and 18-months (12-month timepoint not shown). Scale bar: 10 μm. Inset: 10× magnification. Blue on merge: DAPI. B. Quantification of autophagic flux in female motor cortex neurons. Values are GFP+RFP/RFP only puncta per cell ± sem for n = 6-8 female mice/timepoint (2-way ANOVA). C. Autophagic flux in male motor cortex neurons. Confocal images of the motor cortex of male tf-LC3B mice that were fed regular chow or a high-fat diet for 6- and 18-months (12-month timepoint not shown). Scale bar: 10 μm. Inset: 10× magnification. Blue on merge: DAPI. D. Quantification of autophagic flux in male motor cortex neurons. Values are GFP+RFP/RFP only puncta per cell ± sem for n = 7-8 male mice/timepoint (2-way ANOVA). Statistically significant p values comparing diet effects (#); age effects within each diet (coloured) compared to 6-month time point (*) or comparing 12- and 18-month time points (•).

Disabled autophagy is now recognized as a hallmark of aging [33]. Our data support this idea in the heart, but not the brain (motor cortex neurons) or blood (i.e., PBMCs) where autophagic flux did not decrease with age. In contrast, autophagic flux within PBMCs increased in an age-dependent fashion, which is consistent with our recent findings in humans at risk of developing type-2 diabetes [43]. Intriguingly, white adipose tissue also displayed an age-dependent increase in autophagic flux, which is due to reduced expression of the autophagy inhibitor RUBCN [42]. Since PBMCs are currently being used to measure autophagic flux in humans [43,44], it will be important to understand which peripheral organs, tissues or cell-types their flux is correlated with.

Overall, we highlight the importance of considering sex differences and temporal changes when measuring how autophagy behaves *in vivo*.

## Limitations of this study

A key limitation of this study was the absence of an older age group (>18-months) with lower survival rates, as aging phenotypes in many mouse models remain relatively mild at 18 months. Ratiometric readouts varied depending on tissue complexity or analytical technique used (e.g., puncta per cell [brain], puncta per field of view [heart] and median fluorescence intensity [blood]), meaning that autophagic flux could not be directly compared between tissues. It is plausible that increased autophagic flux in PBMCs might be influenced by age- or diet-related changes to the composition of the PBMC pool (e.g., myeloid skewing) [54]. Critically, by 18-months of age, mice that were fed a high-fat diet were obese [55] and most developed tumors [56]. This makes it difficult to uncouple whether changes in autophagic flux were due to the diet *per se*, obesity or cancer.

The tf-LC3B probe is useful for monitoring autophagic flux in tissues [19,29,30] but, like all experimental models and approaches, it has drawbacks. The RFP signal of tf-LC3B resists lysosomal degradation and likely over-represents autolysosome abundance and therefore autophagic flux [57]. Excluding lipofuscin in the brain reduced the risk of misidentifying fluorescent artifacts as autophagosomes or autolysosomes, but it also meant that autolysosomes containing lipofuscin were excluded. The GFP:RFP ratio compares the relative abundance of tf-LC3B within non-acidic (i.e., autophagosomes) and acidic compartments (i.e., autolysosomes) at the time of sacrifice. It is therefore a readout for the delivery of autophagosomes to lysosomes, a measure which correlates with – but is not strictly – autophagic flux. The tf-LC3B probe can in rare cases be misleading: for instance, when lysosomes are non-functional yet still acidic [58]. Fluorescent signals from tf-LC3B may also come from non-canonical autophagy pathways (e.g., conjugation of ATG8s to single membranes [CASM]) [59]. In some tissues, overexpression of tf-LC3B may lead to autophagy inhibition, such as that seen with GFP-LC3B overexpression [60].

## Materials and methods

### Antibodies

Primary antibodies: ATG2B (rabbit polyclonal, PA5-85718, Thermo Fisher); ATG5 (rabbit monoclonal, CST 12994, Cell Signalling Technology); ATG7 (rabbit monoclonal, CST8558, Cell Signalling Technology); ATG13 (rabbit monoclonal, CST 13273, Cell Signalling Technology); ATG16L1 (rabbit monoclonal, CST 8089, Cell Signalling Technology); BECN1 (mouse monoclonal, SC-48341, Santa Cruz); CD3 (clone: 17A2, rat monoclonal-FITC conjugate, CST 86603, Cell Signalling Technology); CD19 (clone: 1D3, rat monoclonal-PE conjugate, CST 82168, Cell Signalling Technology); CTSD (rabbit monoclonal, CST 74089, Cell Signalling Technology); F4/80 (clone: BM8.1, rat monoclonal-VioletFluor450 conjugate, CST 40781, Cell Signalling Technology); FIP200 (rabbit monoclonal, CST 12436, Cell Signalling Technology); GFAP (chicken polyclonal, ab4674, Abcam); GFP (rabbit polyclonal, A-6455, Thermo Fisher) IBA1 (goat polyclonal, ab48004, Abcam); LC3B (rabbit polyclonal, NB100-2220, Novus Biologicals); LMNB1 (rabbit monoclonal, CST 13435, Cell Signalling Technology); NBR1 (rabbit polyclonal, ab126175, Abcam); NeuN (clone: E4M5P, mouse monoclonal, CST 94403, Cell

Signalling Technology); NDP52 (rabbit polyclonal, Ab68588, Abcam); SQSTM1/p62 (rabbit polyclonal, PM045, MBL Life Science (imaging) and mouse monoclonal, H00008878-M01, Novus Biologicals (Immunoblotting); ULK1 (rabbit monoclonal, CST 8054, Cell Signalling Technology).

Secondary Antibodies: Alexa Fluor-488 donkey anti-goat IgG (705-545-003; Jackson ImmunoResearch); Alexa Fluor 488 donkey anti-mouse IgG (715-545-151; Jackson ImmunoResearch); Alexa Fluor 647-donkey anti-mouse IgG (715-605-150; Jackson ImmunoResearch); Alexa Fluor 647-donkey anti-rabbit IgG (711-605-152; Jackson ImmunoResearch); Cy3-donkey anti-chicken IgY/IgG (703-165-155; Jackson ImmunoResearch).

## Animals

tf-LC3 transgenic mice (C57BL/6-Tg (CAG-RFP/EGFP/Map1lc3b) 1Hill/J; RRID:IMSR_JAX:027139) were kindly provided by Associate Professor Bradley Turner (The Florey Institute, Melbourne, Australia) and a breeding colony was established at the South Australian Health and Medical Research Institute (SAHMRI) following rederivation. Mice were genotyped by genomic PCR of ear notches with the following primers:

F-GFP primer: 5'-CATGGACGAGCTGTACAAGT-3'

R-Map1lc3b: 5'-CACCGTGATCAGGTACAAGGA-3'

F-Internal: 5'-CTAGGCCACAGAATTGAAAGATCT-3'

R-Internal: 5'-GTAGGTGGAAATTCTAGCATCATCC-3'

For this study, 96 tf-LC3 male and female mice (50:50) were randomly assorted to chow (15.6 MJ/Kg, 12.3% total calculated digestible energy from lipids, SF17-091, Specialty Feeds, Australia) or high-fat diet (19.0 MJ/Kg, 43.0% total calculated digestible energy from lipids, SF16−001, Specialty Feeds, Australia) groups from two-months of age. Four additional C57BL/6J (non-transgenic) mice were used as negative controls. Mice were then humanely sacrificed following terminal anesthesia and cardiac puncture after 6-, 12-, and 18-months of age (8 mice/sex/diet at each timepoint). Mice were group-housed (4/cage) in Techniplast individually ventilated cages in a specific and opportunistic pathogen-free facility. Mice were maintained in a controlled temperature room (between 18–24 °C with 45–75% humidity) and a 12 h light/dark cycle (0730–1930 h; UTC+0930). To prevent malocclusion that could be caused by the diet softness, one wood stick was added to every cage. Cage enrichment consisted of two tunnels per cage with sizzle nest and a nestlet. The mice were given *ad libitum* access to food and water. Food consumption was not monitored. Cages were changed twice/week to limit dermatitis that could be caused by the high-fat diet. Food was topped-up twice/week and mice were weighed fortnightly.

All animal experimentation was approved by the SAHMRI Animal Ethics Committee (SAM21-019).

## Tissue collection and preparation

Mice were fasted for 6 h, weighed then humanely killed after 6-, 12- and 18-months of age. Mice were culled in the morning by terminal anesthesia using isoflurane and cardiac puncture. Blood glucose was measured using the CON-TOUR®NEXT blood glucose monitoring system (Ascensia Diabetes Care Australia). Blood was collected into pre-chilled EDTA tubes and stored on ice before PBMC isolation (see below). Brains were dissected, washed in phosphate-buffered saline (PBS), then sagittally dissected into halves and immersed in 10% neutral buffered formalin (Thermofisher Scientific, Fronine, FNNJJ010). Heart (ventricular muscle) was dissected then washed in PBS and immersed in 10% neutral buffered formalin. One-week after fixation in formalin, tissues were immersed in PBS for one-week before processing.

## Confocal microscopy for tf-LC3 mouse tissues

Tissues from the tf-LC3 mice were embedded in 6% low-melt agarose solution (Bio-rad, 1613111) in individual Peel-A-Way embedding cryomolds (Merck) and stored in PBS at 4 °C in the dark until vibratome sections were cut. Tissue slices

(40 µm thick) were prepared with a vibratome (VT1000S, Leica Microsystems) in chilled PBS. Coverslips were mounted onto glass slides with Vectashield antifade mounting media containing DAPI (Vector Laboratories, Abacus DX, H-1200) and sealed with nail polish. Slides were kept at 4 °C in the dark until analysis. Heart and motor cortex sections were imaged at 40× objective with 5× magnification on a Leica TCS SP8X/MP confocal microscope. Images were taken from the same region for each animal for continuity, and focused and imaged using the DAPI channel. The experimenter was blinded to samples throughout imaging and analysis.

## Analysis of tf-LC3B puncta to monitor autophagic flux in tissues

GFP⁺ and RFP⁺ signals from the tf-LC3B probe were quantified in a manner similar to that previously described [61]. Image analysis was performed using FIJI ImageJ.

For the motor cortex, neurons were identified based on DAPI⁺ nucleus morphology (i.e., based their large size, circular shape, unfilled/ "open" lumen and nucleoli-like inclusions) which reliably identified NeuN⁺ neurons (confirmed by receiver operating characteristic curve analysis; AUC = 1) (see below). Lipofuscin was identified and excluded if it was observed in the DAPI channel. Each neuronal cell body was traced and saved in the 'ROI manager'. Images were converted to an 8-bit format and then GFP⁺ and RFP⁺ channels were separated, thresholded, made binary, then puncta were counted within each neuron using the 'analyze particles' function (area = > 0.05 µm²; circularity = 0.00–1.00). GFP⁺ and RFP⁺ puncta (double positive, i.e., autophagosomes) were identified using the 'AND' function of the 'image calculator' then counted within each neuron using the 'analyze particles' function (area = > 0.05 µm²; circularity = 0.00–1.00). GFP⁻ and RFP⁺ puncta (RFP only, i.e., autolysosomes) were obtained by subtracting GFP⁺ and RFP⁺ puncta (double positive, i.e., autophagosomes) from the total number of RFP⁺ puncta (i.e., autophagosomes and autolysosomes). Then a ratio of autophagosomes:autolysosomes per cell was used to estimate autophagic flux whereby a high autophagosomes:autolysosomes ratio indicates that autophagic flux is low, whereas a low autophagosomes:autolysosomes ratio indicates that autophagic flux is high.

Hearts were analyzed in a similar fashion as motor cortex neurons but with minor differences. Lipofuscin was not detected and therefore did not have to be excluded, and cell boundaries were not traced since individual heart muscle cells were typically longer than the field of view. Therefore, a ratio of autophagosomes:autolysosomes per field of view was used to estimate autophagic flux. In some images, sarcomeres were visible in the GFP channel. However, because their signal was diffuse throughout the cell rather than compact like autophagosomes, they did not interfere with the analysis of GFP⁺ puncta after thresholding and particle analysis.

## Immunofluorescence

Fixed tissues were paraffin-embedded for microtome sectioning (6 µm thick sagittal sections, rotary microtome; Leica, RM2235) and mounted upon Superfrost Plus Slides (Thermofisher Scientific, Menzel Glaser, SF41296SP). Paraffin sections were dewaxed and rehydrated in xylene and a graded ethanol series prior to microwave antigen retrieval in a 10 mM citrate, 0.05% Tween 20 (pH 6.0) buffer. Sections were blocked (PBS containing 10% normal donkey serum) for 2 h and incubated overnight with primary antibodies diluted in 2% normal donkey serum at room temperature. After washing in PBS, sections were incubated with species-specific fluorophore-conjugated secondary antibody for 2 h, washed, and then mounted with coverslips using VectaShield with DAPI (Vector Laboratories, Abacus DX, H-1200). Slides were kept at 4 °C in the dark until analysis on a Leica TCS SP8X/MP confocal microscope.

## Analysis of cell-specific markers and nuclei size in the brain

Motor cortex sections were imaged at 40 × objective on a Leica TCS SP8X/MP confocal microscope. Image analysis was performed using QuPath-0.4.4. To test for accuracy of visual identification of neurons using DAPI, the experimenter was first blinded to the cell-type marker channels: neurons (NeuN), microglia (IBA1) and astrocytes (GFAP). Then the DAPI channel was used to identify nuclei that were large, circular, with an unfilled/ "open" lumen with nucleoli-like inclusions

(typical of neurons) which were manually selected using the cell counting tool. Following this, channels corresponding cell-type markers were overlayed to verify "true"- and "false"-positives for neuronal nuclei which were counted. Accuracy and sensitivity of this method was determined by receiver operating characteristic curve analysis which yielded an AUC = 1, suggesting it was robust and that nuclei characteristics (large, circular, with an unfilled/ "open" lumen with nucleoli-like inclusions) are a reliable way to identify neurons.

## PBMC isolation from tf-LC3 transgenic mouse blood and flow cytometry analysis

Blood samples collected from tf-LC3 mice were used to isolate PBMCs. For each sample tube (~1 mL), cold Dulbecco's phosphate-buffered saline (DPBS; Gibco, Thermo Fisher Scientific, 14190136) was added to blood (1:1) and samples were mixed by gently pipetting 3–4 times. Four mL of Lymphoprep (Stemcell Technologies, 07811) was added to 10 mL conical centrifuge tubes and the blood/DPBS mixture was carefully overlaid. These tubes were centrifuged for 30 min at 800 × g at 4°C (with the brake off). PBMCs (white layer at the interface of plasma, upper phase, and Lymphoprep, trans-lucent phase) were carefully aspirated (~2 mL) with a 1 mL pipette and dispensed in a 10 mL conical centrifuge tube. Cold DPBS was added to the PBMCs to a final volume of 6 mL and were mixed gently by inverting tubes 3–4 times. PBMCs were then pelleted by centrifugation at 600 x g for 10 min at 4°C (with brake on). The supernatant was discarded and the cells were resuspended in 2 mL red blood cell lysis buffer (1X, BD Biosciences, 555,899), mixed by pipetting up-and-down, maintained on ice for 5 min, and mixed again. Five mL of cold DPBS was added and the tubes were centrifuged at 600 x g for 5 min at 4°C. The supernatant was discarded and the cells were washed one last time in 5 mL cold DPBS and centrifuged at 600 x g for 5 min at 4°C. The supernatant was discarded and the cells were resuspended in 0.5 mL cold DPBS; the suspension was transferred to a 5 mL tube (Falcon tubes, 5 mL polystyrene; Thermofisher Scientific, 352008) and kept on ice and in the dark before flow cytometry analysis. When required, isolated PBMCs from C57BL/6J (non-transgenic) mice were immunostained for T lymphocyte (CD3+), B lymphocyte (CD19+), and monocyte (F4/80+) markers.

PBMCs were analyzed by flow cytometry on a BD LSR Fortessa X20 Analyser (BD Bioscience, USA). PBMCs were identified based on internal complexity (i.e., granularity; SSC-A) and size (FSC-A) to exclude debris and potential con-tamination by erythrocytes and granulocytes during the PBMC extraction step. SSC-H versus SSC-A were used to isolate single cells and remove doublets from the analysis. Within this population, GFP and RFP florescence intensities were obtained with BD FACSDiva software (BD Bioscience, USA), similar to that previously described [62,63]. PBMCs from C57BL/6J (non-transgenic) mice served as a background control when analyzing tf-LC3B mice. Median fluorescence intensities (per cell) for GFP and RFP within the PBMC pool were used to calculate GFP:RFP using FlowJo software (Tree Star Inc., Ashland, OR, USA). Importantly, to ensure data consistency over time, the analyser was calibrated at the beginning of each day as per manufacturer's instructions.

## Immunoblotting

Hearts were washed in PBS then stored at −80°C. Hearts were homogenised in RIPA lysis buffer, and combined with LDS sample buffer and reducing agent, then proteins were resolved by SDS-PAGE, and transferred onto PVDF membranes. Membranes were blocked for 1 h with 5% skim milk in TBST and probed for indicated proteins overnight at 4°C. Membranes were washed with TBST, probed with HRP-conjugated secondary antibodies for 1 h, re-washed and imaged by chemiluminescence.

## Statistical analysis

Graphs and statistical analyses were generated using Prism 10 (GraphPad Software, La Jolla, CA, USA; version 10.2.3). Normal distribution was assessed with the Shapiro-Wilk normality test. Parametric analyses were used for normally distributed data and non-parametric analyses were used for non-normally distributed data. Figure legends show descrip-tions of data point values, error bars, statistical tests, and sample size. Results were considered significantly different

when p < 0.05. Statistically significant p values comparing diet effects are denoted with # symbol; age effects within each diet compared to the 6-month time point are denoted with * symbol; age effects within each diet comparing 12- and 18-month time points are denoted with • symbol; p < 0.05 = 1 symbol (i.e., #, * or •); p < 0.01 = 2 symbols (i.e., ##, ** or ••); p < 0.001 = 3 symbols (i.e., ###, *** or •••); p < 0.0001 = 4 symbols (i.e., ####, **** or ••••). We started animal time course experiments with a sample size of n = 8 mice/sex/diet/timepoint. However, several mice died throughout the experiment: one mouse died and three had to be ethically culled due to dermatitis (S1D and S1E Fig). This meant that in some cases <8 mice were analyzed. Moreover, PBMCs were analyzed in <8 mice at the 6-month timepoint due to a flow cytometer breakdown.

## Supporting information

**S1 Fig. Metabolic and physiological effects of aging and high-fat diet.** A. Body weight of tf-LC3B mice that were fed regular chow or a high-fat diet. Values are mean body weight (g) ± sem for n = 6–24 mice/sex/diet/timepoint. B. Fasting blood glucose levels of female tf-LC3B mice that were fed regular chow or a high-fat diet. Values are blood mean glucose levels (mmol/L) ± sem for n = 6–8 female mice/diet/timepoint (2-way ANOVA). C. Fasting blood glucose levels of male tf-LC3B mice that were fed regular chow or a high-fat diet. Values are mean blood glucose levels (mmol/L) ± sem for n = 7–8 male mice/diet/timepoint (2-way ANOVA with Tukey's multiple comparisons test). D. Tumor incidence and mortality rates of female tf-LC3B mice that were fed regular chow or a high-fat diet; n values (mice/sex/diet/timepoint) are indicated by phenotype. E. Tumor incidence and mortality rates of male tf-LC3B mice that were fed regular chow or a high-fat diet; n values (mice/sex/diet/timepoint) are indicated by phenotype. F. Survival analysis of female tf-LC3B mice that were fed regular chow or a high-fat diet. n = 8 female mice/diet/timepoint. G. Survival analysis of male tf-LC3B mice that were fed regular chow or a high-fat diet. n = 8 female mice/diet/timepoint. Statistically significant p values comparing diet effects (#); age effects within each diet (coloured) compared to 6-month time point (*) or comparing 12- and 18-month time points (•). (PDF)

**S2 Fig. Verification of PBMCs.** A. PBMCs purified from blood are enriched with monocytes as well as T and B lymphocytes. PBMCs extracted from C57BL/6J (non-transgenic) mice were immunostained for monocyte (F4/80) as well as T- (CD3) and B-lymphocyte (CD19) markers and analyzed via flow cytometry. This illustrates that gating parameters employed were appropriate. B. Variation of autophagic flux in female PBMCs. Values are coefficient of variation in GFP/RFP median fluorescence intensity across cells within each animal. Data is related to Fig 2A. C. Variation of autophagic flux in male PBMCs. Values are coefficient of variation in GFP/RFP median fluorescence intensity across cells within each animal is shown. Data is related to Fig 2B. (PDF)

**S3 Fig. Autophagosome and autolysosome abundance, and autophagic flux variability in the heart.** A. Quantification of autophagosomes in the female heart. Values are GFP + RFP puncta per field ± sem for n = 6–8 female mice/timepoint (2-way ANOVA). B. Quantification of autolysosomes in the female heart. Values are RFP only puncta per field ± sem for n = 6–8 female mice/timepoint (2-way ANOVA). C. Quantification of autophagosomes in the male heart. Values are GFP + RFP puncta per field ± sem for n = 7–8 male mice/timepoint (2-way ANOVA with Tukey's multiple comparisons test). D. Quantification of autolysosomes in the male heart. Values are RFP only puncta per field ± sem for n = 7–8 male mice/timepoint (2-way ANOVA). E. Variation of autophagic flux in the female heart. Coefficient of variation in GFP + RFP/RFP only puncta per field across 10 fields for each animal. Data is related to Fig 3A and B. F. Variation of autophagic flux in the male heart. Coefficient of variation in GFP + RFP/RFP only puncta per field across 10 fields for each animal. Data is related to Fig 3C and D. Statistically significant p values comparing diet effects (#); age effects within each diet (coloured) compared to 6-month time point (*) or comparing 12- and 18-month time points (•). (PDF)

**S4 Fig. Abundance of key autophagy proteins in the heart.** A. Immunoblot analysis of heart lysates from female tf-LC3B mice that were fed regular chow or a high-fat diet. Blots were probed as indicated. n = 2 mice/diet/timepoint is shown. B. Immunoblot analysis of heart lysates from male tf-LC3B mice that were fed regular chow or a high-fat diet. Blots were probed as indicated. n = 2 mice/diet/timepoint is shown. C. Heatmap showing levels of indicated proteins in heart lysates from female tf-LC3B mice that were fed regular chow or a high-fat diet. Quantification of immunoblots in S. Fig 4A. Values are mean for n = 2 mice/diet/timepoint. HMW, high molecular weight. D. Heatmap showing levels of indicated proteins in heart lysates from male tf-LC3B mice that were fed regular chow or a high-fat diet. Quantification of immunoblots in S. Fig 4B. Values are mean for n = 2 mice/diet/timepoint. HMW, high molecular weight.
(PDF)

**S5 Fig. Large DAPI nuclei as a proxy for identifying NeuN+ neurons, and DAPI puncta for detection of lipofuscin in the brain.** A. Immunofluorescence processing quenches GFP and RFP signals. Confocal images of the brain of tf-LC3B mice after processing for immunofluorescence (without primary or secondary antibodies). Scale bar: 10 μm. Inset: 10 × magnification. Blue on merge: DAPI. B. DAPI-stained nuclei morphology of NeuN+ neurons. Confocal images of the motor cortex of female non-transgenic control mice that were fed regular chow for 12-months, immunostained for neuronal (NeuN), astroglial (GFAP), and microglial (IBA1) markers. Scale bar: 30 μm. Inset: 8 × magnification. Blue on merge: DAPI. C. Morphological features of DAPI-stained nuclei (large size, circular shape, unfilled/ "open" lumen, and nucleoli-like inclusions) can reliably distinguish NeuN+ neurons when used as selection criteria by a blinded experimenter. Approximately 1400 DAPI+ nuclei/cells from n = 2 mice/sex were analyzed (receiver operating characteristic curve analysis; AUC = 1). D. Identification of lipofuscin autofluorescence in the brain. Confocal images of the brain of tf-LC3B mice. The image shown is duplicated from (Fig 4A) but highlights evidence of fluorescent artefacts caused by lipofuscin in the inset. Scale bar: 10 μm. Inset: 10 × magnification. Blue on merge: DAPI. E. Identification of lipofuscin autofluorescence in the brain. Confocal images of the brain of non-transgenic C57BL/6 mice. Fluorescent artefacts caused by lipofuscin are displayed in the inset. Scale bar: 10 μm. Inset: 10 × magnification. Blue on merge: DAPI.
(PDF)

**S6 Fig. Autophagosome and autolysosome abundance, and autophagic flux variability in motor cortex neurons.** A. Quantification of autophagosomes in female motor cortex neurons. Values are GFP + RFP puncta per field ± sem for n = 6–8 female mice/timepoint (2-way ANOVA). B. Quantification of autolysosomes in female motor cortex neurons. Values are RFP only puncta per field ± sem for n = 6–8 female mice/timepoint (2-way ANOVA). C. Quantification of autophagosomes in male motor cortex neurons. Values are GFP + RFP puncta per field ± sem for n = 7–8 male mice/timepoint (2-way ANOVA). D. Quantification of autolysosomes in male motor cortex neurons. Values are RFP only puncta per field ± sem for n = 7–8 male mice/timepoint (2-way ANOVA). E. Variation of autophagic flux in female motor cortex neurons. Coefficient of variation in GFP + RFP/RFP only puncta per cell across 10 fields for each animal. Data is related to Fig 4A and B. F. Variation of autophagic flux in male motor cortex neurons. Coefficient of variation in GFP + RFP/RFP only puncta per cell across 10 fields for each animal is shown. Data is related to Fig 4C and D. Statistically significant p values comparing diet effects (#); age effects within each diet (coloured) compared to 6-month time point (*) or comparing 12- and 18-month time points (•).
(PDF)

**S7 Fig. SQSTM1/p62 inclusions within motor cortex neurons.** A. SQSTM1/p62 inclusions in female motor cortex neurons. Confocal images of the brain of female tf-LC3B mice that were fed regular chow or a high-fat diet for 6- and 18-months (12-month timepoint not shown), immunostained for SQSTM1/p62 and the neuronal marker NeuN. Note that our immunofluorescence protocol quenched GFP and RFP signals from tf-LC3B (S3 Fig) allowing those channels to be used for detection. Scale bar: 10 μm. Inset: 10 × magnification. Blue on merge: DAPI. B. Quantification of SQSTM1/p62 inclusions in female brain neurons. Values are SQSTM1+ puncta per NeuN+ neuron for n = 3–8 female mice/timepoint (2-way ANOVA). C. SQSTM1/p62 inclusions in male brain neurons. Confocal images of the brain of male tf-LC3B mice

that were fed regular chow or a high-fat diet for 6- and 18-months (12-month timepoint not shown), immunostained for SQSTM1/p62 and the neuronal marker NeuN. Scale bar: 10 μm. Inset: 10× magnification. Blue on merge: DAPI. D. Quantification of SQSTM1/p62 inclusions in male brain neurons. Values are SQSTM1+ puncta per NeuN+ neuron for n = 4–8 male mice/timepoint (2-way ANOVA). Statistically significant p values comparing diet-effects (#); age-effects within each diet (coloured) compared to 6-month time point (*) or comparing 12- and 18-month time points (•). (PDF)

**S8 Fig. Uncropped immunoblots from S4 Fig.** (PDF)

## Acknowledgments

We thank Dr Sanjna Singh (Lysosomal Health in Ageing at SAHMRI) for experimental advice.

## Author contributions

**Conceptualization:** Julien Bensalem, Timothy J Sargeant.

**Data curation:** Alexis Martin, Leanne K. Hein, Sofia Hassiotis.

**Formal analysis:** Julian M. Carosi.

**Funding acquisition:** Julien Bensalem, Timothy J Sargeant.

**Investigation:** Alexis Martin, Leanne K. Hein, Sofia Hassiotis, Kathryn J. Hattersley, Célia Fourrier.

**Methodology:** Leanne K. Hein, Sofia Hassiotis.

**Project administration:** Timothy J Sargeant.

**Resources:** Bradley J Turner, Timothy J Sargeant.

**Supervision:** Timothy J Sargeant.

**Writing – original draft:** Julian M. Carosi.

**Writing – review & editing:** Julian M. Carosi, Alexis Martin, Leanne K. Hein, Sofia Hassiotis, Kathryn J. Hattersley, Bradley J Turner, Célia Fourrier, Julien Bensalem, Timothy J Sargeant.

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
