## [Decision Letter · Decision Letter 0]

7 Dec 2024

PLOS ONE

Dear Dr. Sargeant,

Thank you for submitting your manuscript to PLOS ONE. After careful consideration, we feel that it has merit but does not fully meet PLOS ONE’s publication criteria as it currently stands. Therefore, we invite you to submit a revised version of the manuscript that addresses the points raised during the review process.

**Reviewer 1**

The authors report aging effects on the aging effects on autophagy flux across three tissues (peripheral blood monocytes, heart, and motor cortex neurons) by examining a tandem-fluorescent LC3B mouse at 6, 12 and 18 months of age. In addition, mice fed a high-fat diet were also evaluated at each group. Both sexes were studied.

The study addresses a significant gap in the field using a novel mouse model and a naturally aging model.

MAJOR COMMENTS:

1. The presentation of the mouse phenotype is incomplete. This aspect is particularly important given the naturally aging model. Specifically, information regarding appearance and weight gain is in Suppl. Fig. 1, but additional data seem necessary (e.g., survival curves or tumor free survival) to better understand the model (some of this data appears to have been collected given the n of animals indicated in Suppl. Fig. 1). Other behavioral and motor performance characteristics would be desirable. Similarly, information on non-transgenic mouse of the same background seems important to understand possible differences in aging effects.

2. The high fat diet mice did not develop fasting hyperglycemia (suppl. Fig. 1) despite apparently greater weight gain at all ages

3. The aging phenotype of many mouse models is very limited by 18 months of age (survival rates are usually >75%) and the lack of an older group with lower survival rates is not justified in the manuscript.

4. The characterization of autophagy flux is limited to the GFP:RFP ratio in cells of interest, but other measurements (e.g., puncta per cell, LC3 abundance are not presented).

5. The presumed DAPI+ of lipofuscin granules is intriguing. It would be expected that pigment granules containing lipofuscin would be cytosolic (mostly lysosomes) and lacking DNA (e.g., PLoS ONE 2024; 19(7): e0306275. https://doi.org/10.1371/journal.pone.0306275). This aspect of the work is insufficiently presented to provide confidence in this methodological approach to address autofluorescence in the brain, particularly in older animals.

6. Many essential methodological aspects of confocal fluorescence imaging (e.g., image acquisition, selection of dynamic range, quantification of puncta) are not provided. These aspects are particularly important for the model characterization and the tissue-based measures. For example, sarcomeres appear visible on some of the GFP images of the heart.

7. Corroborative information would be useful in comparing the measured changes in autophagy flux with other measures of autophagy. For instance, there appears to be very low levels of GFP detected suggesting the near absence of autophagosomes. Whether there are global changes in LC3 protein is not clear.

8. The analyses of GFP:RFP ratio presented in figs. 2-4 only reflect mean values per cell/field, clustered across animals, and there is no information presented on the variability across cells within animals, which may be of interest given expected increases in heterogeneity with older age.

9. Concluding statements regarding aging effects on autophagy flux in the heart and brain (results sections and first paragraph of the Discussion) do not seem to be supported by the figures. There are no significant post-hoc differences highlighted in Fig 3B or 4B/D, even if a main age effect is reported. None of the diet effects were significant in the 2-way ANOVA and a significant interaction with age was only reported in the male heart.

MINOR COMMENTS:

1. The quantification of GFP:RFP ratios is insufficiently presented in the results and figure legends to facilitate interpretation (Fig. 2 uses “median fluorescence intensity”, Figs. 3 and 4 “puncta per field”.

2. Post hoc analysis in suppl. Fig. 1B are not presented despite a significant age effect.

3. Statements in the text regarding tumor frequency seem to be based on only 8 animals per age/diet/sex group (suppl. Fig. 1D, 1E). A tumor-free survival curve is likely obtainable for the aging colony.

4. The section on motor cortex neurons suggests that p62/SQSTM1 puncta were identified, but there is no such data in any of the figures (including supplemental).

**Reviewer 2**

In their manuscript, Carosi et al. analyze autophagy in PBMCs, heart and motor cortex neurons during aging and high-fat diet. For that, they use a transgenic mouse expressing a tandem- fluorescent LC3 reporter (RFP-GFP-LC3), which allows the quantification of autophagic activity. They show that autophagy changes differently across the different tissues analyzed, with differences in response to aging and HFD, and between males and females. The manuscript addresses an interesting topic, and tries to elucidate how autophagy changes during aging and HFD in different tissues. However, there are some concerns the authors should address to add strength to their conclusions.

1. The main issue with the data shown in this manuscript is that the authors have only used the transgenic mouse model expressing RFP-GFP-LC3 reporter. Although very useful, it has some limitations (as the authors state in their conclusions). The activity and acidity of the lysosome are two key aspects in the interpretation of the results obtained using this technique. For example, changes in lysosomal degradation without changes in the pH would lead to the misinterpretation of the results. In this regard, I suggest the authors to evaluate lysosomal activity and pH. Although challenging in tissues in vivo, they can assess it easily in PBMCs using lysosensor and other probes that can be used for flow cytometry.

2. In Figure 2, the authors show autophagic flux in PBMCs isolated from male and female mice at different ages and diet. Although they show the data in PBMCs in general, it would be interesting to show autophagic flux in the different main cell populations ((T cells, B cells, monocytes, etc).

3. The addition of some data regarding the expression of the main autophagy regulators and proteins would add important information and would help to strengthen the conclusions of the study.

We look forward to receiving your revised manuscript.

Kind regards,

Vladimir Trajkovic

Academic Editor

PLOS ONE

Journal Requirements:

This investigation was supported by Lysosomal Health in Ageing at SAHMRI, and an Ideas Grant from the National Health and Medical Research Council (GNT2002608) awarded to JB and TJS. JMC is supported by an EMCR Fellowship from The Hospital Research Foundation Group (2022-CF-EMCR-007). The authors acknowledge Microscopy Australia (ROR: 042mm0k03) resources at the Future Industries Institute, University of South Australia, enabled by NCRIS.  

This investigation was supported by Lysosomal Health in Ageing at SAHMRI, and an Ideas

Grant from the National Health and Medical Research Council (GNT2002608) awarded to JB

and TJS. JMC is supported by an EMCR Fellowship from The Hospital Research Foundation

Group (2022-CF-EMCR-007). The authors acknowledge Microscopy Australia (ROR:

042mm0k03) resources at the Future Industries Institute, University of South Australia,

enabled by NCRIS.

This investigation was supported by Lysosomal Health in Ageing at SAHMRI, and an Ideas Grant from the National Health and Medical Research Council (GNT2002608) awarded to JB and TJS. JMC is supported by an EMCR Fellowship from The Hospital Research Foundation Group (2022-CF-EMCR-007). The authors acknowledge Microscopy Australia (ROR: 042mm0k03) resources at the Future Industries Institute, University of South Australia, enabled by NCRIS.

4. In the online submission form, you indicated that The data underlying the results presented in the study are available from the corresponding author under reasonable request.

Reviewers' comments:

Reviewer's Responses to Questions

**Comments to the Author**

1. Is the manuscript technically sound, and do the data support the conclusions?

Reviewer #1: Yes

Reviewer #2: Partly

2. Has the statistical analysis been performed appropriately and rigorously?

Reviewer #1: Yes

Reviewer #2: Yes

3. Have the authors made all data underlying the findings in their manuscript fully available?

Reviewer #1: Yes

Reviewer #2: Yes

4. Is the manuscript presented in an intelligible fashion and written in standard English?

Reviewer #1: Yes

Reviewer #2: Yes

Reviewer #1: The authors report aging effects on the aging effects on autophagy flux across three tissues (peripheral blood monocytes, heart, and motor cortex neurons) by examining a tandem-fluorescent LC3B mouse at 6, 12 and 18 months of age. In addition, mice fed a high-fat diet were also evaluated at each group. Both sexes were studied.

The study addresses a significant gap in the field using a novel mouse model and a naturally aging model.

MAJOR COMMENTS:

1. The presentation of the mouse phenotype is incomplete. This aspect is particularly important given the naturally aging model. Specifically, information regarding appearance and weight gain is in Suppl. Fig. 1, but additional data seem necessary (e.g., survival curves or tumor free survival) to better understand the model (some of this data appears to have been collected given the n of animals indicated in Suppl. Fig. 1). Other behavioral and motor performance characteristics would be desirable. Similarly, information on non-transgenic mouse of the same background seems important to understand possible differences in aging effects.

2. The high fat diet mice did not develop fasting hyperglycemia (suppl. Fig. 1) despite apparently greater weight gain at all ages

3. The aging phenotype of many mouse models is very limited by 18 months of age (survival rates are usually >75%) and the lack of an older group with lower survival rates is not justified in the manuscript.

4. The characterization of autophagy flux is limited to the GFP:RFP ratio in cells of interest, but other measurements (e.g., puncta per cell, LC3 abundance are not presented).

5. The presumed DAPI+ of lipofuscin granules is intriguing. It would be expected that pigment granules containing lipofuscin would be cytosolic (mostly lysosomes) and lacking DNA (e.g., PLoS ONE 2024; 19(7): e0306275. https://doi.org/10.1371/journal.pone.0306275). This aspect of the work is insufficiently presented to provide confidence in this methodological approach to address autofluorescence in the brain, particularly in older animals.

6. Many essential methodological aspects of confocal fluorescence imaging (e.g., image acquisition, selection of dynamic range, quantification of puncta) are not provided. These aspects are particularly important for the model characterization and the tissue-based measures. For example, sarcomeres appear visible on some of the GFP images of the heart.

7. Corroborative information would be useful in comparing the measured changes in autophagy flux with other measures of autophagy. For instance, there appears to be very low levels of GFP detected suggesting the near absence of autophagosomes. Whether there are global changes in LC3 protein is not clear.

8. The analyses of GFP:RFP ratio presented in figs. 2-4 only reflect mean values per cell/field, clustered across animals, and there is no information presented on the variability across cells within animals, which may be of interest given expected increases in heterogeneity with older age.

9. Concluding statements regarding aging effects on autophagy flux in the heart and brain (results sections and first paragraph of the Discussion) do not seem to be supported by the figures. There are no significant post-hoc differences highlighted in Fig 3B or 4B/D, even if a main age effect is reported. None of the diet effects were significant in the 2-way ANOVA and a significant interaction with age was only reported in the male heart.

MINOR COMMENTS:

1. The quantification of GFP:RFP ratios is insufficiently presented in the results and figure legends to facilitate interpretation (Fig. 2 uses “median fluorescence intensity”, Figs. 3 and 4 “puncta per field”.

2. Post hoc analysis in suppl. Fig. 1B are not presented despite a significant age effect.

3. Statements in the text regarding tumor frequency seem to be based on only 8 animals per age/diet/sex group (suppl. Fig. 1D, 1E). A tumor-free survival curve is likely obtainable for the aging colony.

4. The section on motor cortex neurons suggests that p62/SQSTM1 puncta were identified, but there is no such data in any of the figures (including supplemental).

Reviewer #2: In their manuscript, Carosi et al. analyze autophagy in PBMCs, heart and motor cortex neurons during aging and high-fat diet. For that, they use a transgenic mouse expressing a tandem- fluorescent LC3 reporter (RFP-GFP-LC3), which allows the quantification of autophagic activity. They show that autophagy changes differently across the different tissues analyzed, with differences in response to aging and HFD, and between males and females. The manuscript addresses an interesting topic, and tries to elucidate how autophagy changes during aging and HFD in different tissues. However, there are some concerns the authors should address to add strength to their conclusions.

1. The main issue with the data shown in this manuscript is that the authors have only used the transgenic mouse model expressing RFP-GFP-LC3 reporter. Although very useful, it has some limitations (as the authors state in their conclusions). The activity and acidity of the lysosome are two key aspects in the interpretation of the results obtained using this technique. For example, changes in lysosomal degradation without changes in the pH would lead to the misinterpretation of the results. In this regard, I suggest the authors to evaluate lysosomal activity and pH. Although challenging in tissues in vivo, they can assess it easily in PBMCs using lysosensor and other probes that can be used for flow cytometry.

2. In Figure 2, the authors show autophagic flux in PBMCs isolated from male and female mice at different ages and diet. Although they show the data in PBMCs in general, it would be interesting to show autophagic flux in the different main cell populations ((T cells, B cells, monocytes, etc).

3. The addition of some data regarding the expression of the main autophagy regulators and proteins would add important information and would help to strengthen the conclusions of the study.

**Do you want your identity to be public for this peer review?** For information about this choice, including consent withdrawal, please see our Privacy Policy

Reviewer #1: No

Reviewer #2: No

---

## [Author Response · Author response to Decision Letter 1]

2 Mar 2025

Reviewer 1

The authors report aging effects on the aging effects on autophagy flux across three tissues (peripheral blood monocytes, heart, and motor cortex neurons) by examining a tandem-fluorescent LC3B mouse at 6, 12 and 18 months of age. In addition, mice fed a high-fat diet were also evaluated at each group. Both sexes were studied.

The study addresses a significant gap in the field using a novel mouse model and a naturally aging model.

MAJOR COMMENTS:

1. The presentation of the mouse phenotype is incomplete. This aspect is particularly important given the naturally aging model. Specifically, information regarding appearance and weight gain is in Suppl. Fig. 1, but additional data seem necessary (e.g., survival curves or tumor free survival) to better understand the model (some of this data appears to have been collected given the n of animals indicated in Suppl. Fig. 1). Other behavioral and motor performance characteristics would be desirable. Similarly, information on non-transgenic mouse of the same background seems important to understand possible differences in aging effects.

We have now provided a survival curve for tf-LC3B mice used in this study (Figure S2F and S2G).

We agree that behavioural and motor performance experiments would be desirable, but we believe that they are beyond the scope of this study which seeks to analyse autophagic flux (using a tf-LC3B florescent reporter) during aging or high-fat diet.

Age-matched non-transgenic controls were included for technical purposes (e.g. to control for background) but were not bred in sufficient numbers for thorough investigation of autophagic flux since they do not express the tf-LC3B reporter.

2. The high fat diet mice did not develop fasting hyperglycemia (suppl. Fig. 1) despite apparently greater weight gain at all ages

By 6-months of age, male but not female mice fed a high-fat diet displayed higher fasting blood glucose compared to those fed regular chow (Figure S1C). However, by 12-months of age, male mice fed chow had higher fasting blood glucose than those fed a high-fat diet, and by 18-months of age, fasting blood glucose levels dropped to a similar level regardless of their diet. Therefore, our study shows temporal dynamics of how fasting blood glucose levels change in response to both a regular and high-fat diet.

3. The aging phenotype of many mouse models is very limited by 18 months of age (survival rates are usually >75%) and the lack of an older group with lower survival rates is not justified in the manuscript.

We have now included this point as a limitation in our discussion: “A key limitation of this study was the absence of an older age group (>18-months) with lower survival rates, as aging phenotypes in many mouse models remain relatively mild at 18-months”.

4. The characterization of autophagy flux is limited to the GFP:RFP ratio in cells of interest, but other measurements (e.g., puncta per cell, LC3 abundance are not presented).

We have now included quantification of autophagosomes (GFP+RFP puncta) and autolysosomes (RFP only puncta) for the heart (Figure S3A-D) and motor cortex neurons (Figure S5A-D) which were used to calculate autophagosome/autolysosome ratios (GFP+RFP / RFP only puncta) presented in main figures (Figure 3A-D and 4A-D). We have also provided immunoblot analysis of tf-LC3B (anti-GFP) and endogenous LC3B in the heart (Figure S4A and S4B).

5. The presumed DAPI+ of lipofuscin granules is intriguing. It would be expected that pigment granules containing lipofuscin would be cytosolic (mostly lysosomes) and lacking DNA (e.g., PLoS ONE 2024; 19(7): e0306275. https://doi.org/10.1371/journal.pone.0306275). This aspect of the work is insufficiently presented to provide confidence in this methodological approach to address autofluorescence in the brain, particularly in older animals.

You are correct that lipofuscin lacks DNA. However, due to its broad autofluorescence (excitation: 320–480 nm / emission: 460–630 nm), it can still be detected using the DAPI filter set (PMID: 16455164; PMID: 20072918), even though it is not DAPI-positive. Upon review, we recognized that our original wording was inaccurate and potentially misleading, and we have now corrected the text: “because lipofuscin throughout the brain exhibits broad autofluorescence, we excluded GFP+ or RFP+ puncta that appeared in the DAPI channel from our analysis”.

You also correctly note that lipofuscin is cytoplasmic and can localize to lysosomes. In our study, we adopted a conservative approach when analyzing brain tissue, quantifying only puncta that were not detected using the DAPI filter. This strategy minimized the risk of mistaking fluorescent artifacts for autophagosomes or autolysosomes (Figure S5D and S5E). Our approach was based on recommendations from a study using the GFP-LC3B mouse (PMID: 14699058). However, we acknowledge that this method may lead to an underrepresentation of autolysosomes containing lipofuscin. We have now addressed this limitation in our manuscript: “Excluding lipofuscin in the brain reduced the risk of misidentifying fluorescent artifacts as autophagosomes or autolysosomes, but it also meant that autolysosomes containing lipofuscin were excluded.”

6. Many essential methodological aspects of confocal fluorescence imaging (e.g., image acquisition, selection of dynamic range, quantification of puncta) are not provided. These aspects are particularly important for the model characterization and the tissue-based measures. For example, sarcomeres appear visible on some of the GFP images of the heart.

We have now included additional information relating to image acquisition in the “Confocal microscopy for tf-LC3 mouse tissues” section of methods: “Images were taken from the same region for each animal for continuity, and focused and imaged using the DAPI channel”.

Information relating to quantification of puncta was outlined in detail in the “Analysis of tf-LC3B puncta to monitor autophagic flux in tissues” section of methods.

We did not establish dynamic range for our imaging. However, consistent confocal acquisition settings were used throughout the time course, and images were acquired and analysed under blinded conditions. Since our image analysis converts puncta signals into a binary format, eliminating intensity-related information, dynamic range was less critical in the context of our experiments.

We have now included additional information relating to GFP+ signal associated with heart sarcomeres in the “Analysis of tf-LC3B puncta to monitor autophagic flux in tissues” section of methods: “In some images, sarcomeres were visible in the GFP channel. However, because their signal was diffuse throughout the cell rather than compact like autophagosomes, they did not interfere with the analysis of GFP+ puncta after thresholding and particle analysis”. It is worthwhile to note that others using GFP-LC3B mice also see GFP+ signals associated with sarcomeres when imaging muscle tissues (PMID: 25484088).

7. Corroborative information would be useful in comparing the measured changes in autophagy flux with other measures of autophagy. For instance, there appears to be very low levels of GFP detected suggesting the near absence of autophagosomes. Whether there are global changes in LC3 protein is not clear.

Since autophagy is a dynamic process, static measurements of individual proteins cannot reliably be used to measure autophagic flux (PMID: 33634751). For example, LC3B-II levels alone are uninformative—high LC3B-II levels may result from either increased autophagosome formation or impaired lysosomal turnover, while low LC3B-II levels could reflect reduced autophagosome formation or enhanced degradation by lysosomes.

To corroborate our findings, we have generated protein expression data for key components of the autophagy pathway to offer additional context. We have now provided immunoblots of heart lysates from male and female tf-LC3B mice fed either a chow or high-fat diet over the time course (Figure S4A and S4B). The heart was selected as it is a homogeneous tissue (in terms of cell type) where autophagic flux changes with age (in both sexes) and with a high-fat diet (in males only). The immunoblots were probed for multiple components of the autophagy pathway, including initiation machinery, the VPS34 complex, lipid scramblase, ATG8 conjugation machinery, ATG8 proteins, autophagy receptors, and lysosomal enzymes. However, we found no clear relationship between the expression of these components and autophagic flux.

Our data suggest that when autophagic flux is high, only a few autophagosomes are present (e.g., ~1 autophagosome per field at 6 months of age in the heart). As flux slows with aging, the number of autophagosomes increases (e.g., ~20 per field at 24 months). This low abundance of autophagosomes during high flux is due to rapid delivery to lysosomes. Critically, the low autophagosome abundance we observe in our study is consistent with previous work using GFP-LC3B mice (PMID: 14699058).

Global changes in tf-LC3B levels in chow-fed mice (detected via immunoblot with an anti-GFP antibody) were stable across time points (Figure S4A and S4B). However, we do observe a minor increase in tf-LC3B levels in high-fat diet-fed mice (Figure S4A and S4B).

8. The analyses of GFP:RFP ratio presented in figs. 2-4 only reflect mean values per cell/field, clustered across animals, and there is no information presented on the variability across cells within animals, which may be of interest given expected increases in heterogeneity with older age.

We have now included quantification of variability in autophagic flux for each mouse. We provide a coefficient of variation (CV) for each mouse based on autophagic flux values taken from 10 images per animal for the heart (Figure S3E and S3F) and motor cortex neurons (Figure S6E and S6F), or ~50,000 PBMCs analysed in blood (Figure S2B and S2C).

In the blood, variability in autophagic flux readings is low (CV = ~0.5 - 0.6) and remains stable over time (Figure S2B and S2C).

In the heart, variability in autophagic flux readings is moderate (CV = ~1 - 2) at 6-months of age (i.e., when autophagic flux is high) but decreases to a low level (CV = ~0.5) at later time points (i.e., when autophagic flux is lower) (Figure S3E and S3F).

In female motor cortex neurons, variability in autophagic flux readings is moderate (CV = ~2) and remains stable over time (Figure S6E). In male motor cortex neurons, variability in autophagic flux readings is high (CV = ~4) in chow fed mice and moderate in high-fat diet mice (CV = ~2) at 6 months of age (Figure S6F). Variability in autophagic flux readings reduced with age to moderate levels (CV = ~2) in chow fed mice, but remained relatively stable in those fed a high-fat diet (Figure S6F).

Therefore, variability in autophagic flux readings does not increase with age, but rather decreases or remains stable depending on tissue, diet or sex.

9. Concluding statements regarding aging effects on autophagy flux in the heart and brain (results sections and first paragraph of the Discussion) do not seem to be supported by the figures. There are no significant post-hoc differences highlighted in Fig 3B or 4B/D, even if a main age effect is reported. None of the diet effects were significant in the 2-way ANOVA and a significant interaction with age was only reported in the male heart.

Although there was a main effect for age in the female heart (p<0.0001) (Figure 3B) and brain (p=0.015) (Figure 4B) post-hoc analysis was not performed because age*diet interactions were not significant (p>0.05). We have therefore softened the conclusions and descriptions of these results in the text.

MINOR COMMENTS:

1. The quantification of GFP:RFP ratios is insufficiently presented in the results and figure legends to facilitate interpretation (Fig. 2 uses “median fluorescence intensity”, Figs. 3 and 4 “puncta per field”.

It was necessary to report ratiometric data differently for each tissue due to variations in tissue complexity or analysis method.

PBMCs: Flow cytometry was used to measure GFP and RFP fluorescence intensity per cell. Since flow cytometry is not imaging-based, it was not possible to analyze autophagosome/autolysosome puncta in PBMCs. We have successfully used flow cytometry to measure autophagic flux in this manner previously (PMID: 28641977).

Heart cells: Due to their large size, imaging at a magnification suitable for resolving autophagosomes/autolysosomes made it difficult to capture entire cells within a single field of view. As a result, we analyzed puncta per field of view rather than per cell.

Motor cortex neurons: We identified neurons based on DAPI morphology (Figure S5B and S5C), allowing us to distinguish them from other cell types such as microglia and astroglia. This distinction enabled us to analyze puncta per neuron rather than per field.

We have acknowledged these methodological differences as a limitation in the Discussion section: “Ratiometric readouts varied depending on tissue complexity or analytical technique used (e.g., puncta per cell [brain], puncta per field of view [heart] and median fluorescence intensity [blood]), meaning that autophagic flux could not be directly compared between tissues”.

2. Post hoc analysis in suppl. Fig. 1B are not presented despite a significant age effect.

There was a main effect for age relating to fasting blood glucose levels in females (p<0.0001) (Figure S1B). However, post-hoc analysis was not performed because the age*diet interaction was not significant (p=0.629).

3. Statements in the text regarding tumor frequency seem to be based on only 8 animals per age/diet/sex group (suppl. Fig. 1D, 1E). A tumor-free survival curve is likely obtainable for the aging colony.

As per comment #1 we have now provided a survival curve.

4. The section on motor cortex neurons suggests that p62/SQSTM1 puncta were identified, but there is no such data in any of the figures (including supplemental).

We adjusted the color balance and swapped the pseudocoloring of SQSTM1 (green) and NeuN (red) to enhance the visibility of SQSTM1 puncta (Figure S7A and S7C).

Reviewer 2

In their manuscript, Carosi et al. analyze autophagy in PBMCs, heart and motor cortex neurons during aging and high-fat diet. For that, they use a transgenic mouse expressing a tandem- fluorescent LC3 reporter (RFP-GFP-LC3), which allows the quantification of autophagic activity. They show that autophagy changes differently across the different tissues analyzed, with differences in response to aging and HFD, and between males and females. The manuscript addresses an interesting topic, and tries to elucidate how autophagy changes during aging and HFD in different tissues. However, there are some concerns the authors should address to add strength to their conclusions.

1. The main issue with the data shown in this manuscript is that the authors have only used the transgenic mouse model expressing RFP-GFP-LC3 reporter. Although very useful, it has some limitations (as the authors state in their conclusions). The activity and acidity of the lysosome are two key aspects in the interpretation of the results obtained using this technique. For example, changes in lysosomal degradation without changes in the pH would lead to the misinterpretation of the results. In this regard, I suggest the authors to evaluate lysosomal activity and pH. Although challenging in tissues in vivo, they can assess it easily in PBMCs using lysosensor and other probes that can be used for flow cytometry.

We acknowledge that analyzing PBMCs using lysosomal pH or acidity probes would provide valuable insights in this study. However, addressing age-dependent changes using this approach would require a significant time investment.

To address this concern, we instead analyzed samples collected throughout the time course by monitoring activation of the lysosomal enzyme CTSD. After synthesis, pre-pro-CTSD undergoes cleavage in the ER to remove its signal peptide, forming pro-CTSD (~52 kDa). This proenzyme is then cleaved by CTSL/CTSB, producing the catalytically active mature CTSD (~34 kDa) (PMID: 16567401). The formation of mature CTSD

---

## [Decision Letter · Decision Letter 1]

22 Apr 2025

Dear Dr. Sargeant,

Thank you for submitting your manuscript to PLOS ONE. After careful consideration, we feel that it has merit but does not fully meet PLOS ONE’s publication criteria as it currently stands. Therefore, we invite you to submit a revised version of the manuscript that addresses the points raised during the review process.

We look forward to receiving your revised manuscript.

Kind regards,

Bin Wu, M.D. & Ph.D.

Academic Editor

PLOS ONE

Journal Requirements:

**Additional Editor Comments:**

The authors have succesfully addressed my questions.

However, it would be important to provide quantifications of the WB for autophagic proteins and cathepsin D.

Reviewers' comments:

Reviewer's Responses to Questions

**Comments to the Author**

Reviewer #2: (No Response)

2. Is the manuscript technically sound, and do the data support the conclusions?

Reviewer #2: Yes

3. Has the statistical analysis been performed appropriately and rigorously?

Reviewer #2: Yes

4. Have the authors made all data underlying the findings in their manuscript fully available?

Reviewer #2: Yes

5. Is the manuscript presented in an intelligible fashion and written in standard English?

Reviewer #2: Yes

Reviewer #2: The authors have succesfully addressed my questions.

However, it would be important to provide quantifications of the WB for autophagic proteins and cathepsin D.

**Do you want your identity to be public for this peer review?** For information about this choice, including consent withdrawal, please see our Privacy Policy

Reviewer #2: No

---

## [Author Response · Author response to Decision Letter 2]

11 May 2025

Response to reviewers:

We have now revised the paper in response to the editor/reviewer #2’s comment: “it would be important to provide quantifications of the WB for autophagic proteins and cathepsin D”.

We have provided quantifications of protein levels from immunoblots in Supplementary Figure 4A and B as heatmaps in Supplementary Figure 4C and D.

---

## [Editor Report · Decision Letter 2]

14 May 2025

Autophagy across tissues of aging mice

PONE-D-24-49236R2

Dear Dr. Sargeant,

We’re pleased to inform you that your manuscript has been judged scientifically suitable for publication and will be formally accepted for publication once it meets all outstanding technical requirements.

Kind regards,

Bin Wu, M.D. & Ph.D.

Academic Editor

PLOS ONE
---

## [Editor Report · Acceptance letter]

PONE-D-24-49236R2

PLOS ONE

Dear Dr. Sargeant,

I'm pleased to inform you that your manuscript has been deemed suitable for publication in PLOS ONE. Congratulations! Your manuscript is now being handed over to our production team.

Kind regards,

on behalf of

Professor Bin Wu

Academic Editor

PLOS ONE